# Inhibition of the mevalonate pathway enhances cancer cell oncolysis mediated by M1 virus

Jiankai Liang[1], Li Guo[1], Kai Li[2], Xiao Xiao[3], Wenbo Zhu[1], Xiaoke Zheng[4], Jun Hu[5], Haipeng Zhang[1], Jing Cai[1], Yaya Yu[1], Yaqian Tan[1], Chuntao Li[1], Xincheng Liu[1], Cheng Hu[6], Ying Liu[7], Pengxin Qiu[1], Xingwen Su[1], Songmin He[8], Yuan Lin[1,9] & Guangmei Yan[1]

Oncolytic virus is an attractive anticancer agent that selectively lyses cancer through targeting cancer cells rather than normal cells. Although M1 virus is effective against several cancer types, certain cancer cells present low sensitivity to it. Here we identified that most of the components in the cholesterol biosynthesis pathway are downregulated after M1 virus infection. Further functional studies illustrate that mevalonate/protein farnesylation/ras homolog family member Q (RHOQ) axis inhibits M1 virus replication. Further transcriptome analysis shows that RHOQ knockdown obviously suppresses Rab GTPase and ATP-mediated membrane transporter system, which may mediate the antiviral effect of RHOQ. Based on this, inhibition of the above pathway significantly enhances the anticancer potency of M1 virus in vitro, in vivo, and ex vivo. Our research provides an intriguing strategy for the rational combination of M1 virus with farnesyl transferase inhibitors to enhance therapeutic efficacy.

[1] Department of Pharmacology, Zhongshan School of Medicine, Sun Yat-sen University, Guangzhou 510080, China. [2] Guangdong Provincial Key Laboratory of Colorectal and Pelvic Floor Disease, The Sixth Affiliated Hospital of Sun Yat-sen University, Guangzhou 510655, China. [3] Department of Pharmacy, The Third Affiliated Hospital of Sun Yat-sen University, Guangzhou 510080, China. [4] Department of Pathology, The First Affiliated Hospital of Sun Yat-sen University, Guangzhou 510080, China. [5] Department of Microbiology, Zhongshan School of Medicine, Sun Yat-sen University, Guangzhou 510080, China. [6] Department of Urology, The Third Affiliated Hospital of Sun Yat-sen University, Guangzhou 510080, China. [7] Department of Infectious Disease, The Third Affiliated Hospital of Sun Yat-sen University, Guangzhou 510080, China. [8] Appel Alzheimer's Disease Research Institute, Brain and Mind Research Institute, Weill Cornell Medical College, New York, New York 10021, USA. [9] Department of Medical Statistics and Epidemiology, School of Public Health, Sun Yat-sen University, Guangzhou 510080, China. These authors contributed equally: Jiankai Liang, Li Guo. Correspondence and requests for materials should be addressed to Y.L. (email: liny96@mail.sysu.edu.cn) or to G.Y. (email: ygm@mail.sysu.edu.cn)

Oncolytic virotherapy has long been considered an attractive option for the treatment of several human cancers, complementing traditional cancer therapies. As a multi-targeting therapy, oncolytic virus possesses cancer cell lytic capacity, leading to innate and adaptive anti-tumor immunity, and it may also specifically lyse tumor vessels to indirectly destroy tumors[1].

M1 is an enveloped Getah-like alphavirus with an 11.7 kb positive single-stranded RNA genome[2]. Genomic and sub-genomic RNA functions as messenger RNA templates for four non-structural and five structural viral proteins[3]. Our previous study showed that M1 is a potent oncolytic virus that is safe and selective towards cancer cells. The replication of the M1 virus in cancer cells causes irresolvable endoplasmic reticulum (ER) stress, which triggers cancer cell apoptosis[4, 5]. However, some cancer cell lines exhibit resistance or low sensitivity to the M1 virus, indicating that the antiviral mechanism in these cancer cells may still effectively combat the replication of the M1 virus[6]. Thus, efforts to dissect the antiviral system in M1 virus-resistant/low sensitive cancer cells and identify effective targets so that we can broaden the therapeutic scope of the M1 virus are still needed.

Lipids are important bioactive molecules that are extremely relevant to virus infection either by providing structural ingredients or by regulating the intracellular signaling pathway[7–9]. As reported, virus infection commonly perturbs host lipid metabolism. For instance, dengue virus or hepatitis C virus (HCV) induces dramatic lipid alterations in fatty acids, sphingolipids, phospholipids, and cholesterol[10, 11]. Notably, several other viruses, such as measles, human immunodeficiency virus (HIV) and West Nile virus, also change the expression of genes in the cholesterol pathway[12, 13]. However, changes in lipid metabolism are either induced by virus to support virus replication or are mediated by the host to suppress it. For example, cholesterol biosynthesis-related genes are suppressed after murine cytomegalovirus infection, which is mediated by the host antiviral factor interferon-β (IFN-β)[14]. Thus, we propose that it is possible to identify a pharmaceutical target to enhance the efficacy of the oncolytic virus by evaluating the changes in lipid levels after virus infection and confirming their roles in virus replication.

To dissect the relationship between lipid metabolism and M1 replication, we first analyzed the expression profiles of lipid-related genes. Most of the cholesterol biosynthesis-related genes are decreased after M1 infection. Further functional studies have revealed that the mevalonate/protein farnesylation/RHOQ axis has an antiviral role during M1 replication in refractory tumor cells. From a therapeutic perspective, we exploited tipifarnib, a highly selective inhibitor of farnesyl transferase (FT) that has passed phase I clinical trials, to efficiently potentiate the oncolytic effect of the M1 virus in vitro, in vivo, and ex vivo. These findings delineate the antiviral effect of the mevalonate pathway on oncolytic virus M1 in refractory cancer cells and offer a mechanism-based combination strategy for potentiated oncolytic virotherapy.

## Results

**Members of the cholesterol biosynthesis pathway are down-regulated after M1 virus infection.** To illustrate the interaction between host lipid metabolism and M1 infection, we first performed expression profiling to detect lipid-related genes in three cancer cell lines, HCT-116, SW1990, and Hep3B. Both HCT-116 and SW1990 are relatively refractory to the M1 virus compared with the sensitive cancer cell line Hep3B (Fig. 1a). Three cell lines were mock infected or infected with the M1 virus; 24 h later, RNA was collected for microarray gene expression profiling. Genes related to four main lipid classes that are important for the

enveloped virus from the Reactome database—fatty acids, phospholipids, sphingolipids, and cholesterol—were analyzed (Supplementary Tables 1-4). Notably, more than 60% of genes related to the cholesterol biosynthesis pathway were downregulated after M1 virus infection in both refractory cell lines, whereas other lipid classes were changed to a limited extent (Fig. 1b–e). In contrast, the metabolism of four lipids were dramatically changed after M1 virus infection in sensitive cancer cell line Hep3B, as the unchanged gene group was obviously decreased, but cholesterol biosynthesis pathway did not show consistent decrease with refractory cells (Supplementary Fig. 1).

To independently validate the microarray data above, quantitative reverse-transcriptase PCR was performed in both cell lines to detect the expression of five selected genes at the beginning of the cholesterol biosynthesis pathway. The mRNA expression of these five genes, HMGCS1, HMGCR, MVD, MVK, and FDPS, was consistently decreased after M1 virus infection in both HCT-116 and SW1990 cell lines (Fig. 1f, g). These results confirmed that cholesterol lipid metabolism pathway was significantly suppressed after M1 virus infection in refractory cancer cells, which indicated its critical role in the propagation of the M1 virus in cancer cells.

**The mevalonate–protein farnesylation branch has an antiviral effect against the M1 virus.** Previous data have indicated that the cholesterol biosynthesis pathway was significantly decreased after M1 virus infection. Here we further explored the functional implication of this phenomenon. Cholesterol biosynthesis involves four stages and the synthesis of mevalonate by β-hydroxy-β-methyl glutaryl-CoA (HMG-CoA) reductase is the first and rate-limiting step of the process. In the second step, the bioactive molecule isoprene is synthesized and provided to synthesize geranyl pyrophosphate, farnesyl pyrophosphate, and squalene in the third step. The fourth step is the transformation of squalene into the steroid core. At the same time, the intermediate geranyl pyrophosphate is supplied for geranylgeranylation type I and II, and farnesyl pyrophosphate is supplied for protein farnesylation (Supplementary Fig. 2)[15].

The observation that the cholesterol biosynthesis pathway is significantly suppressed after M1 virus infection prompted us to further explore the functional implication of this phenomenon. We first used small interfering RNA (siRNA) to knock down HMGCR, the rate-limiting step of cholesterol biosynthesis. M1 virus infection caused a significant reduction in the cell viability of HCT-116 cells after HMGCR knockdown compared with the scrambled control (Fig. 2a). Accordingly, both M1 virus replication and its viral proteins were highly elevated when HMGCR expression was silenced in HCT-116 and SW1990 cells (Fig. 2b–d). Similar results were also observed using HMG-CoA inhibitors, lovastatin, or fluvastatin in HCT-116 cells (Fig. 2e). To further confirm the antiviral function of the cholesterol biosynthesis pathway, the key intermediate farnesyl pyrophosphate (FPP) was added back to the culture system after fluvastatin treatment to rescue the pathway. Replenishment of FPP can reverse M1 virus infection after inhibition of the cholesterol biosynthesis pathway by fluvastatin in both HCT-116 and SW1990, but not in Hep3B (Fig. 2f and Supplementary Fig. 3). These results confirmed the inhibitory effect of the cholesterol biosynthesis pathway on the replication and oncolysis of the M1 virus.

We next sought to identify the downstream branches of the cholesterol biosynthesis pathway that may mediate the above antiviral effect. Using siRNA targeting SQLE specific for squalene epoxidase in cholesterol synthesis, FNTB for FT, PGGT1B for geranylgeranylation type I, and RABGGTB for geranylgeranylation type II, we found that the M1 virus protein expression levels

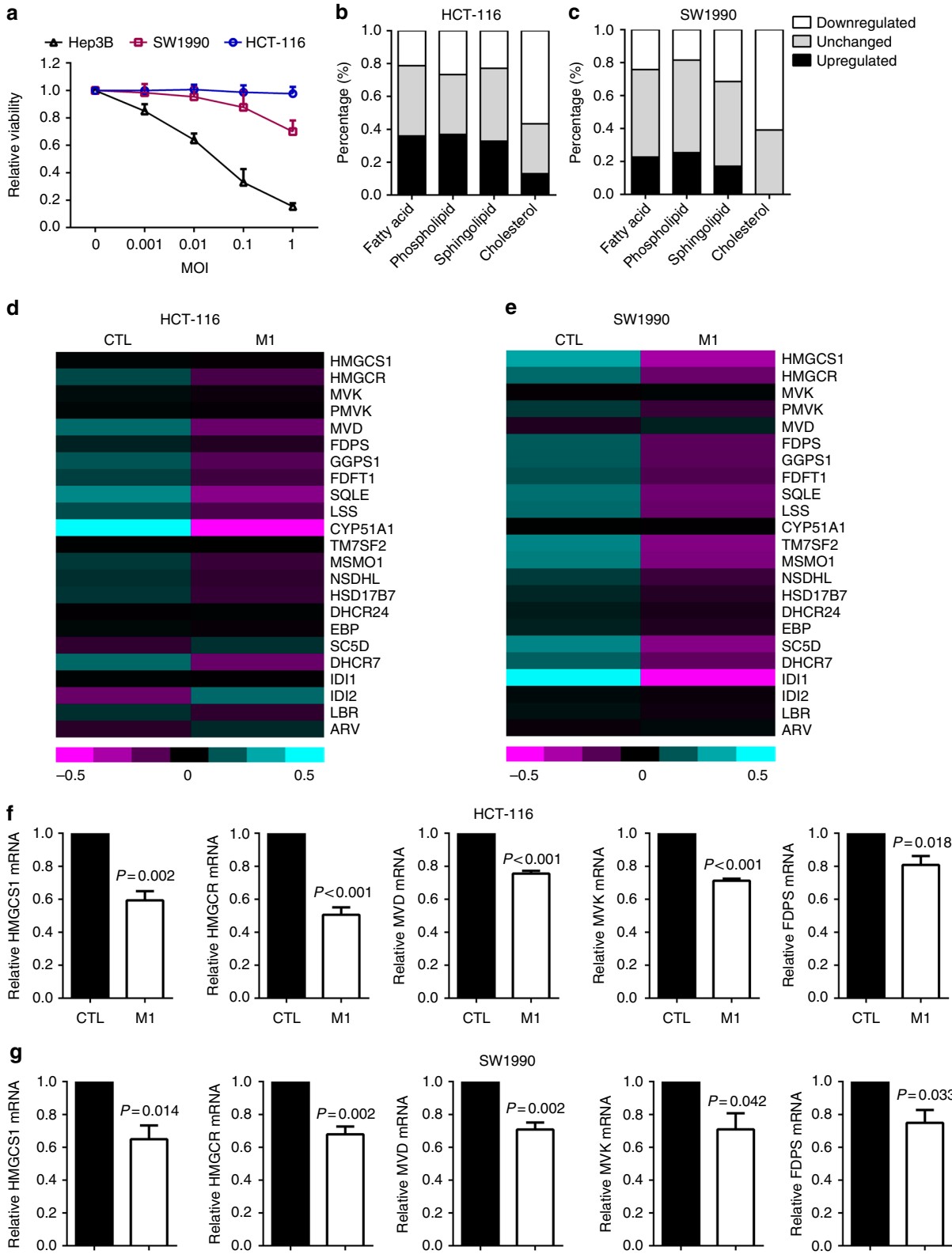

**Fig. 1** Mevalonate–cholesterol synthesis pathway is downregulated after M1 virus infection. **a** HCT-116, SW1990, and Hep3B cells were infected with M1 virus (MOI = 0, 0.001, 0.01, 0.1, 1), and cell viability was determined with MTT assay 72 h after M1 virus infection. $n = 3$. Data shown in **a**, **f**, **g** were the mean ± SEM. **b**, **c** Analyzation of four kinds of lipid-related gene expression changes according to microarray data. HCT-116 and SW1990 cells were treated with M1 virus (MOI = 1) or vehicle. 24 h later, RNA was collected and analyzed by GeneChip Human Genome U133 Plus 2.0 Array (Affymetrix). Microarray analysis was performed on one sample. **d**, **e** Shown is heat maps of the mevalonate-cholesterol pathway related genes expression after M1 virus infection, according to microarray data above. **f**, **g** qRT-PCR detection of mevalonate-cholesterol pathway-related gene expressions after M1 virus infection (MOI = 1) for 24 h in HCT-116 and SW1990. $n = 3$. Statistical significance was using $t$-test, two-sided

were enhanced accordingly after *FNTB* knockdown in both the HCT-116 and SW1990 cancer cell lines but not in the normal cell line L02 (Fig. 2g), suggesting that the protein farnesylation branch is the potential branch mediating antiviral function.

Further confirmation of the protein farnesylation branch using additional siRNAs targeting *FNTB* revealed that M1 virus protein expressions were enhanced as expected, but no apparent difference was observed in the normal cell line L02 after *FNTB* knockdown (Fig. 2h). Meanwhile, M1 virus-induced cell viability loss was enhanced (Fig. 2i, j) and cell apoptosis rate was elevated after *FNTB* knockdown (Supplementary Fig. 4). Accordingly, the virus titer was significantly promoted in both cancer cell lines (Fig. 2k, l). In addition, using small-molecule inhibitors FTI277 and GGTI-2133, which target FT and geranylgeranyl transferase,

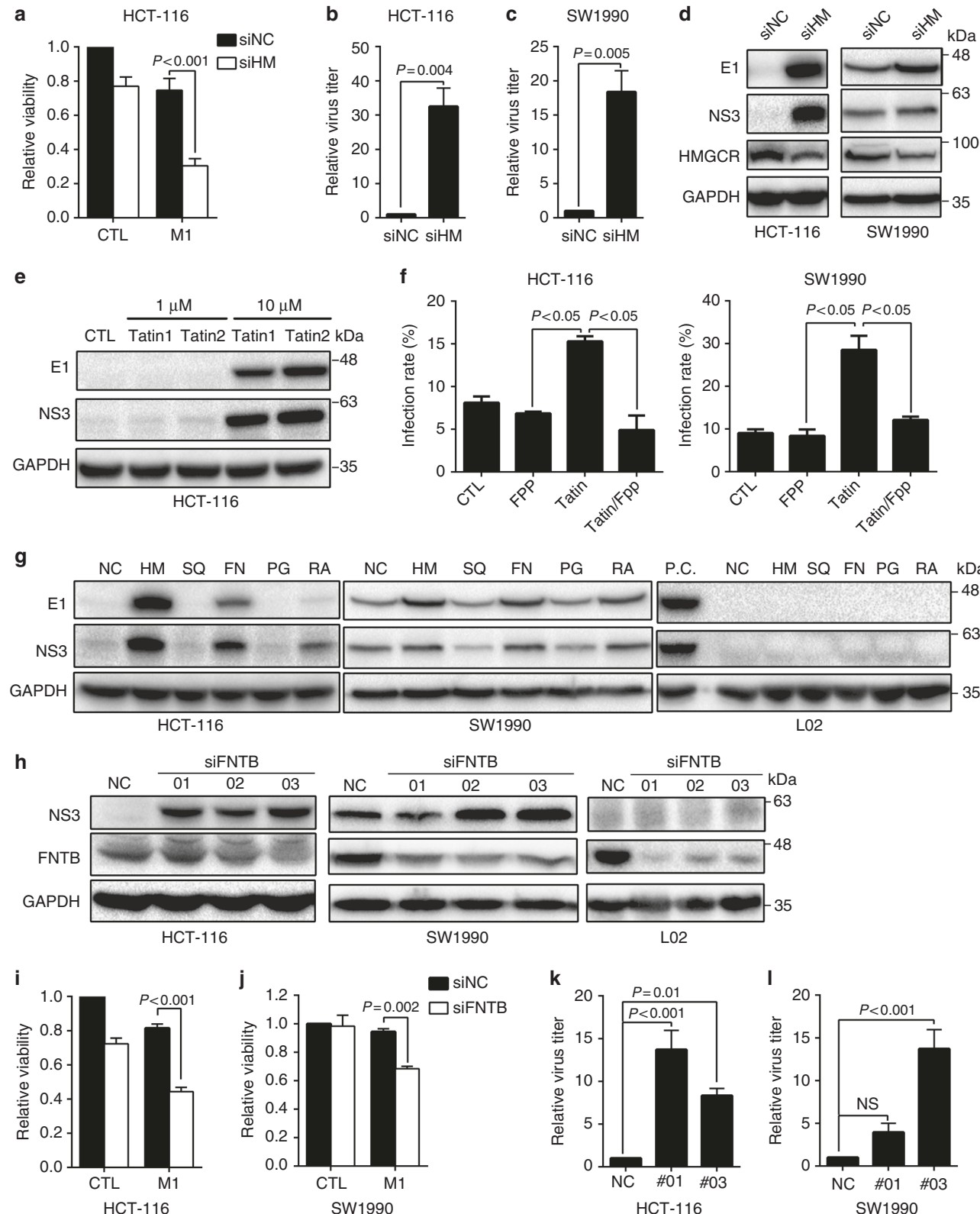

respectively, we found that the inhibition of FT could enhance M1 virus replication and subsequent oncolysis, whereas the inhibition of geranylgeranyl transferase caused little change (Supplementary Fig. 5).

With the above data, we proved that the mevalonate-protein farnesylation branch has an antiviral role in M1 virus infection and suppresses its oncolytic effect.

**siRNA screening indicates *RHOQ* as a farnesylated factor fostering antiviral function.** As a universal posttranslational modification, protein farnesylation is catalyzed by FT to add a lipophilic C15 farnesyl group to the cysteine residue in the C terminus of the protein to facilitate protein membrane association[16]. We sought to determine the specific proteins regulated by FT that are responsible for the antiviral function of protein farnesylation. To identify the farnesylated antiviral factors, we first searched for FT-regulating candidates in Swiss-Prot database (Supplementary Table 5). Next, we screened out the genes that have been reported to be functionally related to the virus, immunity, or small GTPase, because most of the farnesylated substrates are small GTPases. By knocking down gene expression and then testing for M1 virus-induced decrease in cell viability, we further narrowed the candidate pool for verification (Fig. 3a). With this strategy, 18 genes were selected for functional screening to identify their antiviral function and 5 of them clearly enhanced M1 virus-induced cell viability loss after siRNA knockdown (Fig. 3b). Further verification by detecting M1 virus infection, we found that siRNAs of *GNG11*, *RHOQ*, *PTP4A3*, and *NRAS* promoted M1 virus infection, and siRNA targeting *RHOQ* showed the most obvious capacity (Fig. 3b, c and Supplementary Fig. 6).

Combining the viability and infection results, we chose *RHOQ* for further investigation regarding its antiviral role. Phase-contrast microscopy also showed that the M1 virus caused an obvious cytopathic effect after *RHOQ* knockdown (Fig. 3d). The aforementioned data showed that the inhibition of protein farnesylation selectively promoted M1 virus replication in cancer cells rather than in normal cells. Here we further showed that knocking down *RHOQ* presented a similar pattern. The expression levels of the M1 virus proteins and virus titer were significantly elevated in the cancer cell line after *RHOQ* knockdown but not in the normal cell line (Fig. 3e, f). Further validating the antiviral effect of *RHOQ* in another cancer cell line SW1990 by siRNAs, the virus infection, virus-induced cell dead, and virus titer were consistently elevated (Supplementary Fig. 7). These results confirmed that *RHOQ* restricts M1 virus replication.

Next, we investigated whether *RHOQ* mediates the antiviral effect of the protein farnesylation pathway. In accordance with the above results, we found that *RHOQ* was highly expressed in HCT-116 and Capan-1 cells, which are refractory to the M1 virus but was deficient in the M1 virus-sensitive Hep3B and T24 cell lines (Fig. 3g). Moreover, the FT inhibitor tipifarnib notably increased the M1 virus-induced loss of viability in *RHOQ* normal cells in a dose-dependent manner but not in *RHOQ*-deleted cells (Fig. 3h). These results further verified the antiviral effect of *RHOQ* and suggested its substantial role in mediating the antiviral function of the mevalonate/protein farnesylation pathway.

It was reported that *RHOQ* is a small GTPase whose intracellular location and correct function are regulated by protein posttranslational farnesylation[17]. Indeed, using confocal microscopy, we found that *RHOQ* was normally located in the cytoplasm in an aggregated manner, whereas suppression of FT activity by the highly selective FT inhibitor tipifarnib apparently induced *RHOQ* dispersion in cytoplasm (Supplementary Fig. 8), suggesting that *RHOQ* was farnesylated under the physiologic state. Considering that blockade of FT by siRNAs or molecular inhibitors significantly promoted the replication of the M1 virus, we deduced that farnesylation may be essential for the antiviral activity of *RHOQ*. Interestingly, we also observed using confocal microscopy that *RHOQ* expression was downregulated after FT inhibition (Supplementary Fig. 8). Furthermore, we confirmed that *RHOQ* expression was decreased after FT inhibition or combining with the M1 virus via western blotting (Fig. 3i), indicating that FT inhibits M1 virus replication by affecting both the location and expression of *RHOQ*.

***RHOQ* knockdown suppressed Rab GTPase and ATP-mediated membrane transporter system.** The interferon pathway has an important antiviral role during virus infection. When pattern recognition receptor (PRR) recognizes pathogen-associated molecular pattern, the corresponding signaling pathway would be activated to induce IFN-β excretion, followed by antiviral interferon-stimulated gene induction to promote the antiviral status of host cells[18]. It was reported that protein farnesylation inhibitors can decrease inflammatory factors, including IFN-β[19, 20]. Thus, to further explore the antiviral mechanism of the mevalonate/protein farnesylation/*RHOQ* axis, we focused on the regulation of the IFN-mediated antiviral process. First, we attempted to confirm that the blockade of protein farnesylation could inhibit the antiviral function of IFN-β. By detecting six important factors in the IFN-β pathway, we found that pretreatment with the FT inhibitor significantly suppressed the IFN-β-induced upregulation of *IRF7*, *MDA5*, and IFN-β mRNA expression (Supplementary Fig. 9). Moreover, tipifarnib inhibited M1 virus-induced *IRF3* and *IRF7* mRNA expression, and notably promoted M1 virus replication as expected (Supplementary

**Fig. 2** Mevalonate-protein farnesylation branch has an antiviral effect. **a** HCT-116 cells were treated with negative control (NC) or HMGCR (HM) siRNA for 24 h and infected with M1 virus (MOI = 1). Cell viability was determined 72 h after M1 virus infection. *n* = 3. Data shown in **a–c**, **f**, **i–l** were the mean ± SEM. Statistical significance was using one-way ANOVA. **b**, **c** M1 virus titer was measured after NC or HM siRNA pretreatment for 24 h and M1 virus infection (MOI = 0.1) for another 36 h. *n* = 3. Statistical significance was using *t*-test, two-sided. **d** M1 virus proteins, E1 and NS3, were detected after NC or HM siRNA pretreatment for 24 h and M1 virus infection (MOI = 1) for another 24 h. *n* = 2. **e** HCT-116 were treated with Tatin1 (Lovastatin) and Tatin2 (Fluvastatin) (1, 10 μM) or vehicle, and were infected with M1 virus (MOI = 1) for 24 h. M1 virus proteins were detected. *n* = 2. **f** HCT-116 and SW1990 cells were treated with Tatin (Fluvastatin) (5 μM) and FPP was added into the cells simultaneously at concentration of 30 μM. Then cells were infected with M1-GFP virus (MOI = 1) for 24 h. M1 virus infection rate was detected by flow cytometry assay. *n* = 3. Statistical significance was using Kruskal–Wallis test. **g** M1 virus proteins were detected after NC or siRNA pretreatment for 24 h and M1 virus infection (MOI = 1) for another 24 h. Five siRNA included HM (HMGCR), SQ (SQLE), FN (FNTB), PG (PGGT1B), RA (RABGGTB). P.C., positive control. *n* = 2. **h** M1 Virus proteins were detected after NC or three FNTB siRNAs pretreatment for 24 h and M1 virus infection (MOI = 1) for another 24 h. *n* = 3. **i**, **j** HCT-116 and SW1990 cells were treated with negative control (NC) or one FNTB siRNA for 24 h, and infected with M1 virus (MOI = 1). Cell viability was determined 72 h after M1 virus infection. *n* = 3. Statistical significance was using one-way ANOVA. **k**, **l** M1 virus titer was measured after NC or FNTB siRNA pretreatment for 24 h and M1 virus infection (MOI = 1) for another 36 h. *n* = 3. Statistical significance was using one-way ANOVA. NS, no significance

Fig. 10). Further knockdown of *IRF3* or *IRF7* enhanced M1 virus infection in HCT-116 cells (Supplementary Fig. 11). The above data proved that *IRF3* and *IRF7* were required for protein farnesylation-mediated antiviral function. Therefore, we proposed that *RHOQ* regulated by protein farnesylation may exert its antiviral role through *IRF3* and *IRF7*. Unexpectedly, knocking down of *RHOQ* by siRNA facilitated M1 virus replication, but did not suppress *IRF3* and *IRF7* expression (Supplementary Fig. 12).

These results demonstrated that *RHOQ* may not exert its antiviral function through the antiviral factors *IRF3* and *IRF7*.

To find out the antiviral mechanism of *RHOQ*, we further used microarray profiling to analysis the cellular process changes after *RHOQ* deficiency. Analysis of genes significantly changed after *RHOQ* knockdown revealed enrichment in limited pathway, which includes guanosine diphosphate (GDP) binding, trans-membrane movement or transport of substance, response to

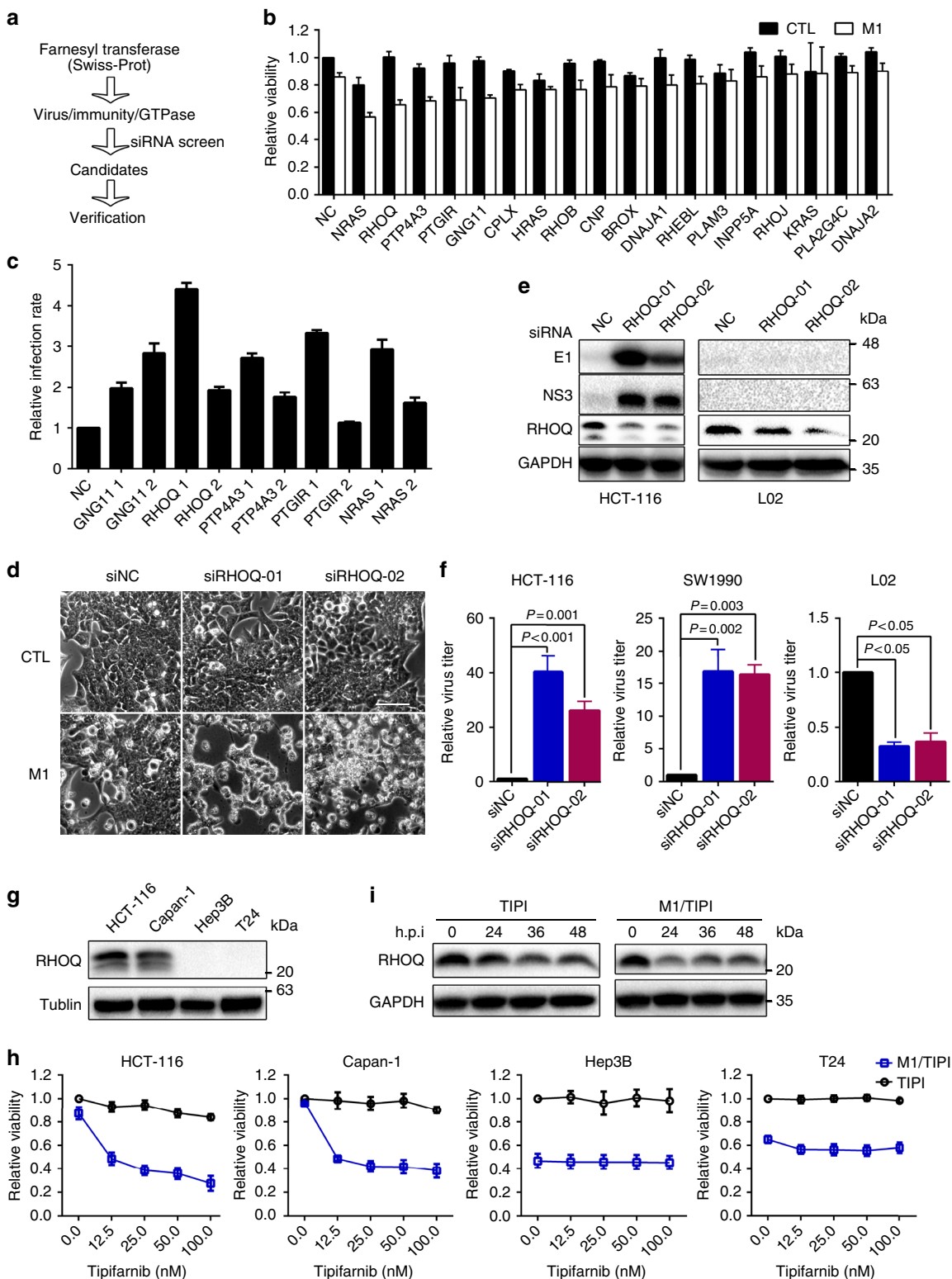

osmotic stress, and phosphatidylinositol biosynthetic process (Fig. 4a, b). Molecular function analysis showed that *RHOQ* knockdown obviously suppressed genes related to Rab GTPase and ATP-mediated membrane transporter system (vacuolar ATPase (V-ATPase) and ATP-binding cassette (ABC) transporters) (Fig. 4c, d), which was closely related to intracellular membrane trafficking and substance transportation. The above data implicated that *RHOQ* was closely involved in the cellular process that were regulated by Rab GTPase, V-ATPase, and ABC transporters, which may mediate the antiviral effect of *RHOQ*.

**The FT inhibitor potentiates the oncolytic efficacy of M1 virus**. As we demonstrated that the inhibition of mevalonate/farnesylation/*RHOQ* pathway selectively promotes the replication of the M1 virus in cancer cells, we proposed that blocking this pathway may substantially potentiate the therapeutic efficacy of the M1 virus. Highly selective FT inhibitors, including tipifarnib and lonafarnib, appear to be good options for the above combination strategy, in that they have been explored to treat cancer patients in phase II clinical trials. Therefore, we tested the combined effect of M1 and tipifarnib in different cell lines. By screening 18 cancer cell lines from 6 cancer types, we concluded that the M1 virus combined with tipifarnib significantly increased cancer cell death in more than 60% of the cell lines. The enhanced oncolysis was most obvious in colon and pancreas cancers, while the liver and brain cancer response was modest, and breast and bladder cancers represented the weakest reaction (Fig. 5a). More importantly, the enhanced oncolysis occurred only in cancer cells but spared normal cells, indicating the safety of this combination strategy (Fig. 5b). Consistent with the above results, tipifarnib promoted virus replication in HCT-116 cancer cells but had no effect in the normal cell line L02 (Fig. 5c).

A previous study showed that the M1 virus induces cancer cell apoptosis. Hence, we investigated whether the combination of M1 and tipifarnib could promote cancer cell death via apoptosis. The Hoechst 33342 staining assay showed that combining the M1 virus with tipifarnib induced a large amount of karyopyknosis, a marker of cell apoptosis, in HCT-116 cancer cells, whereas neither the M1 virus nor tipifarnib alone caused obvious cell karyopyknosis (Fig. 5d). Moreover, the combination M1 virus and tipifarnib treatment significantly enhanced the level of an apoptosis executor, cleaved caspase-3, in sensitive cancer cell lines (Fig. 5e). Finally, we exploited Annexin V/PI staining to confirm that the M1 virus and tipifarnib combination significantly elevated the number of apoptotic cells compared with the M1 virus or tipifarnib monotreatments (Supplementary Fig. 13). These data demonstrated that the M1 virus combined with tipifarnib promotes cancer cell apoptosis.

Given that tipifarnib remarkably potentiates the selective oncolysis of the M1 virus in vitro, we investigated the in vivo antitumor activity. First, we established subcutaneous xenograft nude mouse models using two cancer cell lines, HCT-116 and SW1990. The drug treatment regimens were administered to tumor-bearing mice as shown in Fig. 6a and the M1 virus was intravenously injected. The tumor growth curves were recorded during the experiments; finally, the mice were killed and the tumors were dissociated and recorded. Either the M1 virus or tipifarnib treatment alone suppressed tumor growth slightly, whereas the combination therapy showed significantly greater anticancer efficiency in both xenograft models (Fig. 6b). At the end point of the experiments, we also observed that the tumor volumes of the combination group were substantially smaller than those in the single-treatment groups or the control group (Fig. 6c).

For the above results demonstrating that the M1 virus is specifically elevated by tipifarnib in cancer cells rather than normal cells, we further examined whether the replication of the M1 virus is selectively enhanced in tumor tissue in vivo. By detecting the virus genome, we confirmed that tipifarnib caused a more than 3000-fold increase of the M1 virus in tumor tissue, and replication was highly enriched (Fig. 6d). Mechanistically, we also observed that antiviral factors *IRF3* and *IRF7* were suppressed after the M1 virus and tipifarnib combination treatment (Fig. 6e, f). Finally, we detected the tumor inhibition efficacy of this combination strategy in clinical samples. Ex vivo results showed that combining tipifarnib and M1 virus significantly promoted liver and colorectal tumor cell death (Fig. 6g, h).

In conclusion, these data illustrated that the FT inhibitor plus the M1 virus combination cooperatively suppresses cancer cells in vitro, in vivo and ex vivo. This combination treatment can selectively enhance M1 virus replication and cause subsequent oncolysis in cancer cells.

## Discussion

The discovery that the mevalonate/farnesylation/*RHOQ* pathway, a side branch of the cholesterol biosynthesis pathway, exerts an antiviral effect on the M1 virus in refractory cancer cells but not normal cells urged us to investigate the combination therapy of tipifarnib with the M1 virus. As expected, targeting the above-mentioned pathway can enhance the replication of the M1 virus in cancer cells, thus selectively promoting the therapeutic effect of the M1 virus in cancers.

The mevalonate pathway plays an important role during virus infection via its downstream intermediates, including those involved in cholesterol synthesis, protein geranylgeranylation and protein farnesylation. For example, statins, inhibitors of the mevalonate pathway, inhibit the replication of HCV and HIV by suppressing protein geranylgeranylation; the geranylgeranylation of FBL2 or Rab11a GTPase promotes HCV or HIV replication, respectively[21, 22]. However, in our results, the mevalonate pathway had a controversial role to inhibit the M1 virus by the farnesylation of RHOQ and other genes distinct from protein geranylgeranylation. A direct antiviral functional study of protein farnesylation has not been reported, but some interferon-

**Fig. 3** Small interfering RNA screen indicates RHOQ as a farnesylated factor fostering antiviral function. **a** Schematic of screening strategy to find out farnesylated anti-viral factors. **b** HCT-116 cells were treated with negative control (NC) or siRNAs for 24 h, and then infected with M1 virus (MOI = 1). Cell viability was determined 72 h after M1 virus infection. n = 3. Data shown in **b, c, f, h** were the mean ± SEM. **c** HCT-116 cells were treated with negative control (NC) or siRNAs for 24 h, and then infected with M1-GFP virus (MOI = 1) for 24 h. M1 virus infection rate was detected by flow cytometry assay. GFP expression rate of each gene was normalized to NC. n = 3. **d** Phase-contrast microscope images of HCT-116 cells after NC or RHOQ siRNAs pretreatment for 24 h and M1 virus infection (MOI = 1) for another 48 h. n = 2. Scale bar: 100 μm. **e** M1 virus proteins were detected by western blotting after NC or RHOQ siRNAs pretreatment for 24 h and M1 virus infection (MOI = 1) for another 24 h in HCT-116 and L02 cells. n = 2. **f** M1 virus titer was measured by TCID50 assay after NC or RHOQ siRNA pretreatment for 24 h and M1 virus infection (MOI = 1) for another 48 h in HCT-116, SW1990, and L02 cells. n = 3. Statistical significance was using one-way ANVOA. **g** RHOQ protein expression in four cell lines was detected, including Hep3B, T24, HCT-116, and SW1990 cells. n = 3. **h** Hep3B, T24, HCT-116, and SW1990 cells were treated with vehicle or Tipifarnib (12.5, 25, 50, 100 nM) and mock-infected or infected with M1 virus (MOI = 1). Cell viability was determined 72 h after M1 virus infection. n = 3. **i** RHOQ protein was detected after vehicle or M1 virus infection (MOI = 1) for 24, 36, and 48 h in HCT-116. n = 3

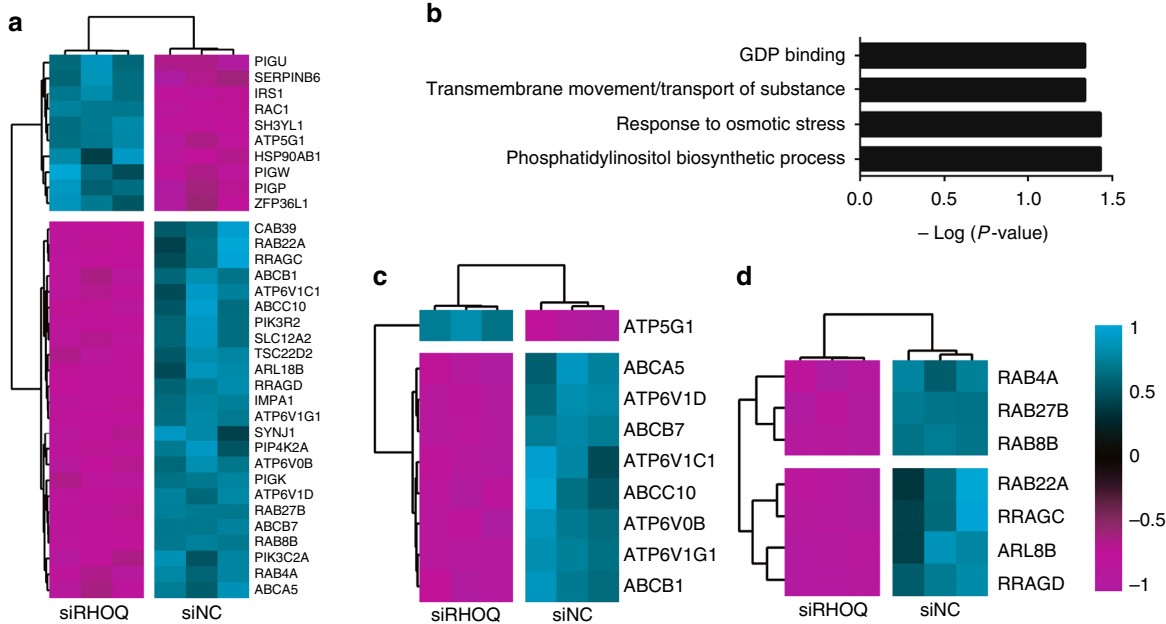

**Fig. 4** RHOQ knockdown suppressed Rab GTPase and ATP-mediated membrane transporter system. HCT-116 were transfected with NC or RHOQ siRNAs for 36 h, and then RNA was collected and analyzed by GeneChip Human Genome U133 Plus 2.0 Array (Affymetrix). Microarray analysis was performed on three biological samples. **a** Heat maps of the gene expressions after RHOQ knockdown; **b** GO analysis of genes in F; **c** heat maps of the V-ATPase family; **d** heat maps of the GDP-binding proteins

stimulated factors reported thus far are regulated by this post-translational modification, such as GBP1, which is not involved in M1 virus infection[23]. We conclude that the four branches under the mevalonate pathway may mediate different functions in virus infection. For M1 virus, the data indicate that protein farnesylation may dominate the antiviral role of this pathway. Thus, specifically blocking the protein farnesylation pathway would guarantee the efficacy of this therapy.

To further explore the antiviral mechanism of *RHOQ*, our data proved that *RHOQ* knock-down did not suppress *IRF3* and *IRF7* expression as FT did. But transcriptome analysis indicated that *RHOQ* regulated V-ATPase, ABC transporters, and Rab GTPases, which were confirmed to regulate intracellular membrane trafficking. In accordance with previous report, *RHOQ* was involved in transmission of neurotransmitter and glucose uptake[24, 25]. Furthermore, both V-ATPase and RAB GTPases are closely relevant to virus infection. V-ATPase is located within intracellular membranes and functions in membrane-trafficking processes such as receptor-mediated endocytosis and intracellular trafficking of lysosomal enzymes by acidification of endocytic compartments[26]. Enveloped viruses, such as influenza virus and vesicular stomatitis virus, enter cells via acidic endosomal compartments, and V-ATPase-induced low PH triggers the formation of a membrane pore through which the viral mRNA can be translocated into the cytoplasm[27]. Besides virus entry, V-ATPase participated in the formation of functional lysosome, which was reported to promote virus entry or hijack viruses by autophagy[28, 29]. Rab GTPases constitute the largest family of small GTPases and have an important role in promoting/inhibiting membrane traffic by regulating vesicle budding, uncoating, motility and fusion[30]. It's clearly demonstrated that viruses use the pre-existing properties of endosomal vesicles to deliver their genetic material to the appropriate site in the cell[31]. Thus, we speculate that *RHOQ* may inhibit the entry and elimination process of M1 virus in the early stage of infection through V-ATPase and

RAB GTPases, and more efforts are still needed to demonstrate the mechanism of *RHOQ* inhibiting M1 virus.

As a multi-target anticancer agent, oncolytic virus is becoming a new drug class for cancer treatment. The first oncolytic virus, T-vec, approved by the Food and Drug Administration, was shown to improve the durable response rate for patients with advanced melanoma[32]. In addition, previous clinical trials of various oncolytic viruses have demonstrated the safety of this therapy. However, most current-generation viruses in clinical trials have failed to meet the efficacy expectations that are set by the preclinical animal model[33]. Further efforts are needed to improve the therapeutic efficacy and the use of a combination therapy with current available cancer therapy is a simple and practical strategy. Drug combination therapy is commonly used in the clinic and studies on oncolytic virus combined with small-molecule drugs have also been reported[34–36]. Combination therapy with clinically used drugs or candidate drugs that have entered phase II or III clinical trials may achieve greater efficacy and safety because those drugs have been testified clinically. Presently, oncolytic combination therapy is mainly based on two mechanisms. First, without enhancing virus replication, small molecules promote cancer cell death by strengthening virus-induced cancer cell stresses, such as ER stress, or inducing the bystander killing effect. This strategy is safe, because a stronger cytopathic effect is induced without enhancing virus replication. However, it remains questionable whether the enhanced cell death will block the spread of oncolytic virus, an activity that may hinder the treatment of cancer clinically. In addition, tipifarnib belongs to the group of oncolytic enhancers, such as dbcAMP and H89, which act through promoting virus replication. Tipifarnib is superior to those molecules, because it has been clinically tested for safety. By repressing the antiviral system of cancer cells, enhancers can promote virus replication and inhibit tumor growth. Considering the characterization of different enhancers, it is reasonable to propose that combining

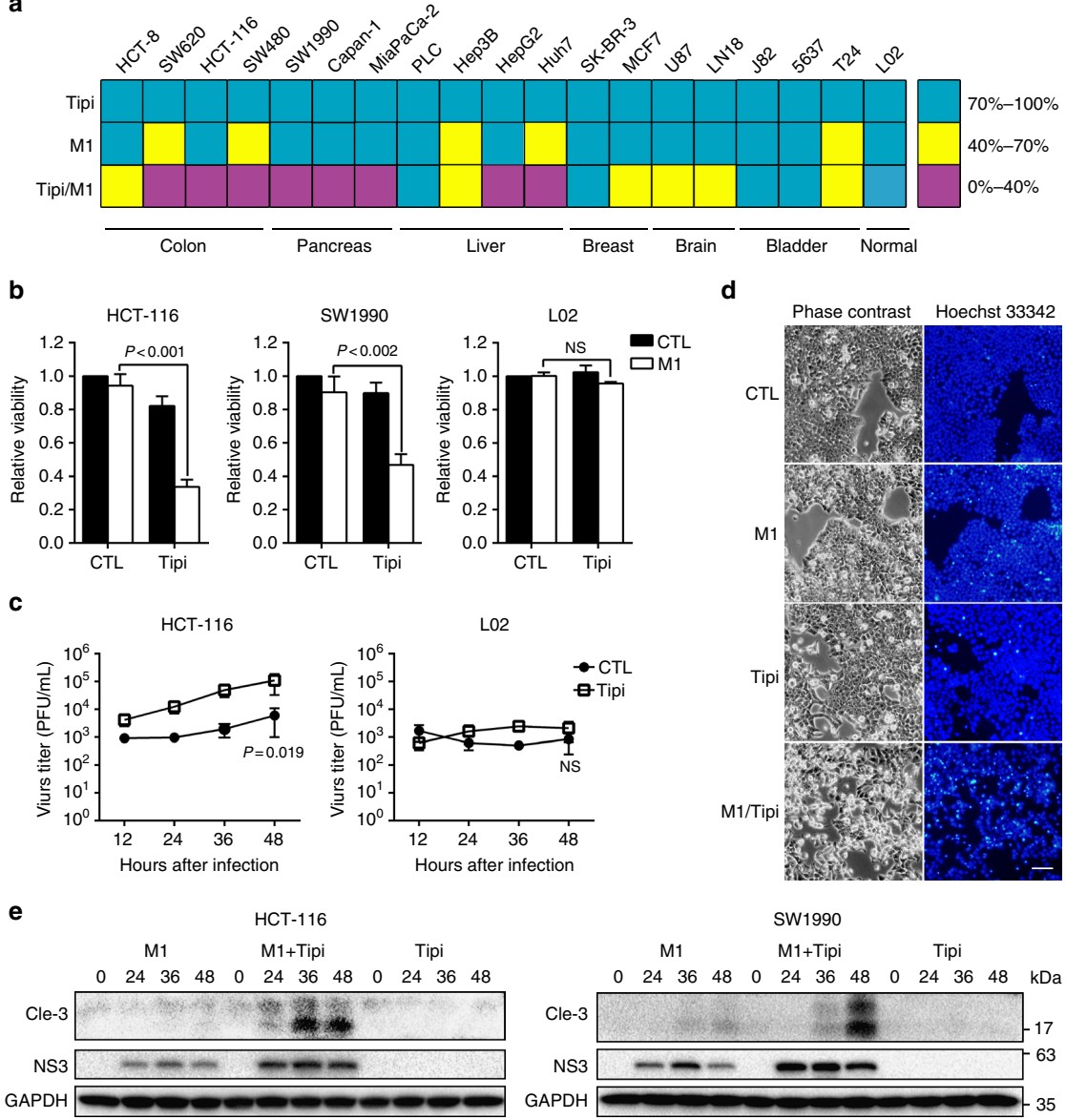

**Fig. 5** Farnesyl transferase inhibitor potentiates the oncolytic efficacy of M1 virus in vitro. **a** Cells were treated with vehicle, Tipifarnib (50 nM), M1 (MOI = 1), or Tipifarnib /M1. Cell viability was determined with MTT assay 72 h after M1 virus infection. $n = 2$. **b** HCT-116, SW1990, and L02 cells were treated with vehicle, Tipifarnib (50 nM), M1 (MOI = 1), or Tipifarnib /M1. Cell viability was determined with MTT assay 72 h after M1 virus infection. $n = 3$. Data shown in **b**, **c** were the mean ± SEM. Statistical significance was using one-way ANOVA. n.s. represents no significance. **c** HCT-116 and L02 cells were treated with vehicle or Tipifarnib (50 nM), and infected with M1 virus (MOI = 0.1). Virus supernatants are collected 12, 24, 36, 48 h after infection. Virus titer of samples was measured by TCID50 assay. $n = 4$. Statistical significance was using $t$-test, two-sided. **d** HCT-116 cells were treated with vehicle, Tipifarnib (50 nM), M1 (MOI = 1), or Tipifarnib /M1 for 48 h, Hoechst 33342 staining was used to detect nucleus. Scale bar: 100 μm. $n = 2$. **e** HCT-116 and SW1990 cells were treated with Tipifarnib (50 nM), M1 (MOI = 1), or Tipifarnib /M1 for 24, 36, 48 h. M1 Virus protein NS3 and apoptosis executor cleaved-caspase3 (Cle-3) were detected by western blotting. $n = 2$

these two enhancers with oncolytic virus may markedly strengthen the anticancer effect.

In addition to the efficacy consideration, combination therapy should also guarantee the safety of the oncolytic virus. We found that the promotion of M1 virus replication and its oncolytic effect through the inhibition of this pathway is cancer selective, leaving normal cells intact. It is commonly accepted that tumor cells are defective in their antiviral system compared with normal cells, a notion that is exploited by oncolytic virus to selectively replicate in and destroy tumor cells[37]. With a more comprehensive antiviral system, we suspect that blocking one of them, the mevalonate/farnesylation/*RHOQ* pathway,

would cause less devastating consequences to the antiviral system in normal cells, in contrast to cancer cells with a weak antiviral system. In a previous study, we showed that the antiviral factor ZAP determines the propagation of the M1 virus, and ZAP expression is lower in cancer cells than in normal cells. This report supports that the anti-M1 factor is weaker in cancers than in normal cells, leaving an opportunity for combination therapy.

Our research focused on the relationship between oncolytic virus and lipid metabolism. By dissecting the cholesterol biosynthesis pathway, we uncovered FT as a new target to promote the efficiency of oncolytic virus M1.

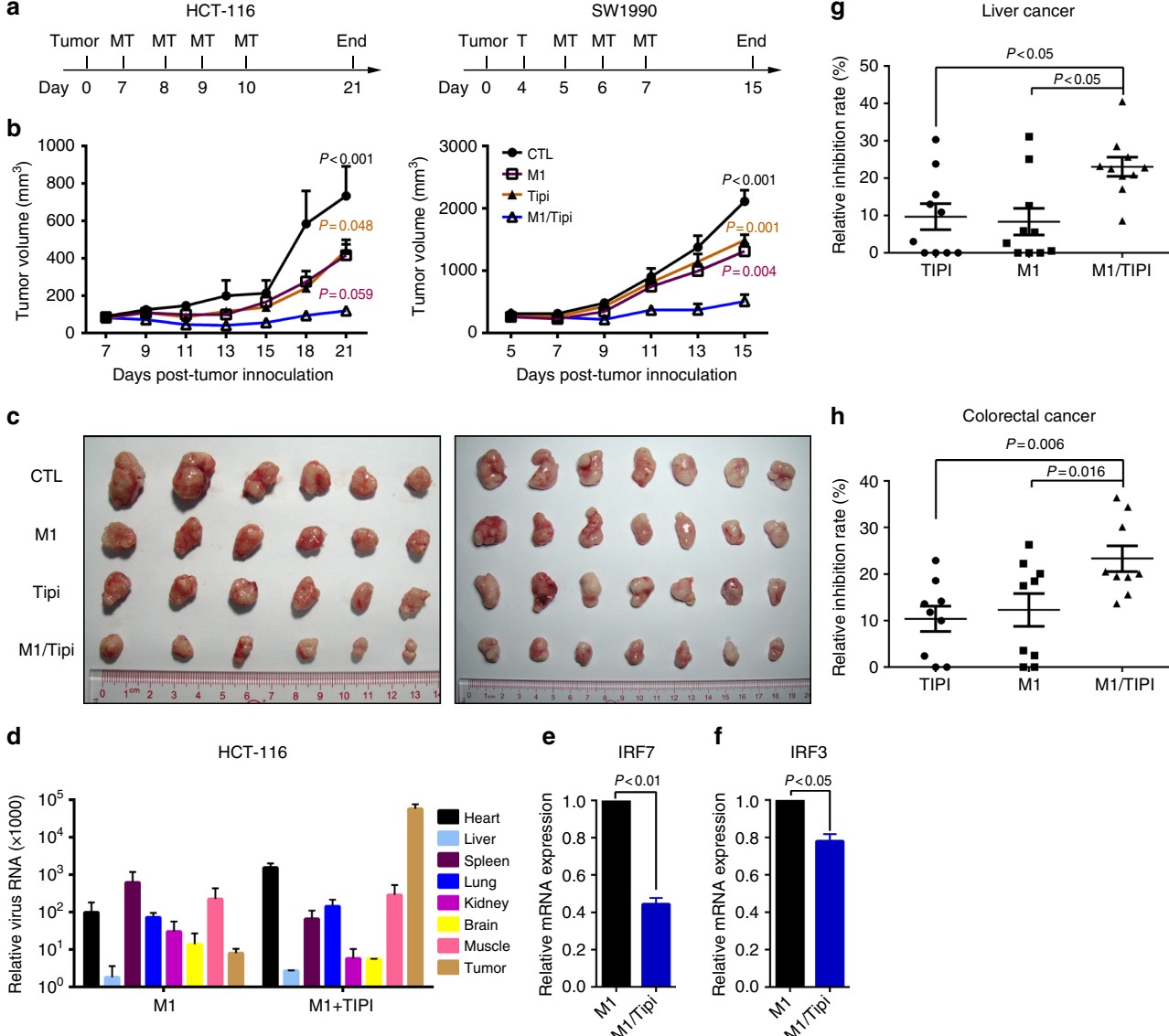

**Fig. 6** Farnesyl transferase inhibitor potentiates the oncolytic efficacy of M1 virus in vivo and ex vivo. **a** Timeline of experimental setup for **b**, **c**. M1 virus (M), Tipifarnib (T), M1 virus/ Tipifarnib (MT). **b**, **c** HCT-116 and SW1990 xenografts were treated with vehicle, Tipifarnib, M1 virus, or Tipifarnib /M1 virus. Tumor growth was assessed by tumor volume measurement over time **b**. At the endpoints, mice were anesthetized and killed. Tumors were subsequently dissected and photographed **c**. Representative data is from four groups, eight mice for each group. Data shown in **b**, **d**, **h** were the mean ± SEM. Statistical significance was using repeated measure ANOVA. Multiple comparisons versus M1/Tipi Group used *LSD-t*. **d** HCT-116 xenografts were treated with M1 virus or Tipifarnib /M1 virus for 3 days. Then mice were killed and main tissues including the heart, muscles, liver, spleen, lung, kidney, brain, and tumor are dissected to detect M1 virus gene expression by qRT-PCR. $n = 3$. **e**, **f** HCT-116 xenografts were treated with M1 virus or Tipifarnib /M1 virus for 3 days. Then mice were killed and tumors are dissected to detect interferon-stimulated genes IRF7 **e** and IRF3 **f** expression by qRT-PCR. $n = 3$. Statistical significance was using *t*-test, two-sided. **g**, **h** Human liver and colorectal tumor tissues were treated with vehicle, tipifarnib (50 nM), M1 ($2 \times 10^6$ PFUs), or a combination for 96 h, and cell viability was assessed by MTT assay. Each point represents one patient. Statistical significance was using one-way ANOVA

## Methods

**Cell culture and viruses**. Cell lines were purchased from the American Type Culture Collection (Maryland, USA), and none are listed in the database of commonly misidentified cell lines maintained by ICLAC (Version 8.0). The cell lines have been authenticated by the short tandem repeat (STR) assay and were confirmed to be without mycoplasma contamination. Cells were cultured in Dulbecco's modifiedd Eagle's medium supplemented with 10% (vol/vol) fetal bovine serum and 1% penicillin/streptomycin. The M1 virus was grown in Vero cells and collected for experiments. The M1 virus titer was determined by the TCID50 assay using BHK-21 cells and the data were converted to PFU.

**Cell viability assay**. Cells were seeded in 96-well plates at 3000 cells per well. After M1 virus treatment, 3-(4,5-dimethylthiazol-2-yl)-2,5-diphenyltetrazolium bromide (MTT) was added (1 mg/ml) and the cells were allowed to grow at 37 °C for 3 h.

The supernatants were removed and the MTT precipitate was dissolved in 100 μl of dimethyl sulfoxide. The optical absorbance was determined at 570 nm by a microplate reader (iMark; Bio-Rad).

**RNA interference**. siRNAs were synthesized by Ribobio (Guangzhou, China). Cells are seeded in proper density and incubated for 24 h, then siRNAs were transfected with Lipofectamine RNAiMAX (13778-150, Thermo Fisher, Rockford, IL, USA) and OPTI-MEM (31985070, Thermo Fisher) in concentration recommended by instruction. Twenty-four hours later, supernatants were removed and changed to new medium. Then the cells were treated according to different experiments. The concentration of siRNA for different genes are as follows: HMGCR (25 nM), FNTB (3 nM for HCT-116, 10 nM for SW1990 and L02), SQLE (40 nM), PGGT1B (40 nM), RABGGTB (40 nM), RHOQ (25 nM for HCT-116, 40 nM for SW1990 and L02), GNG11 (25 nM), PTP4A3 (25 nM), PTGIR (25 nM), NRAS (25 nM), and IRF3 (40 nM), IRF7 (40 nM).

**Antibodies and reagents**. The following antibodies were used in this study: HMGCR (ab174830, Abcam, Shanghai, China, 1:2500); TC10 (AB32079, Abcam, 1:2000); FNTB (3283-1, epitomics, 1:10,000); cleaved caspase-3 (9664 s, Cell Signaling Technology, Danvers, MA, USA, 1:1000); Caspase-3 (9662, Cell Signaling Technology, 1:1000); GAPDH (AP0060; Bioworld, MN, USA, 1:2000); M1 E1 and NS3 (produced by Beijing Protein Innovation, Beijing, China, 1:2000); Tipifarnib (192185-72-1, Selleckchem, Houston, TX, USA); Lovastatin (S2061, Selleckchem); Fluvastatin (S1909, Selleckchem); FTI277 (F9803, Sigma, Saint Louis, MO, USA); and FITC Annexin V Apoptosis Detection Kit I (556547, BD Pharmingen, San Jose, CA, USA).

**Western blot analyses**. Cell samples were prepared using M-PER Mammalian Protein Extraction Reagent (Thermo Fisher) followed by SDS-polyacrylamide gel electrophoresis. The membranes were visualized with a ChemiDoc XRS + System (Bio-Rad) using Immobilon Western Chemiluminescent HRP Substrate (Millipore). Uncropped western blot images of data shown in Figs. 2–5 and supplementary can be found in Supplementary Figures 14, 15.

**Quantitative reverse-transcriptase PCR**. RNA was extracted with TRIzol reagent (Thermo Fisher) and reverse transcription was performed with oligo(dT) and RevertAid Reverse Transcriptase (Thermo Fisher) according to the manufacturer's instructions. Quantitative PCR was performed with SuperReal PreMix SYBR Green (TIANGEN) in the Applied Biosystems 7500 Fast Real-Time PCR System (Life Technologies). The gene expression levels were normalized to those of β-actin and TBP-1. The amplification primers (Thermo Fisher) are listed in Supplementary Table 6.

**Laser-scanning confocal microscopy**. HCT-116 cells were seeded in chambers and were treated or mock treated with tipifarnib (50 nM) for 24 h. The samples were fixed with 5% (v/v) paraformaldehyde, permeabilized with 0.5% (v/v) Triton X-100/phosphate-buffered saline for 7 min and treated with primary antibody (rabbit monoclonal anti-TC10 antibody, 1:500, Abcam) for at least 24 h. After secondary antibody (Alexa Fluor® 555 anti-rabbit IgG, Thermo Fisher, A31572, 1:500) treatment at 37 °C for 1 h, 4',6-diamidino-2-phenylindole was added to the system for the next 15 min. Finally, the samples were observed by laser-scanning confocal microscopy (Nikon).

**Animal models**. This study was approved by Ethics Committee of ZSSOM on Laboratory Animal Care. We estimated to set eight mice for each group. HCT-116 ($5 \times 10^6$ cells/mouse) or SW1990 ($5 \times 10^6$ cells/mouse) cells were inoculated subcutaneously into the hind flank of 4-week-old female BALB/c-nu/nu mice to establish a subcutaneous xenograft model. After palpable tumors developed (50 mm³), the mice were divided into four groups randomly by table of random numbers. M1 virus ($2 \times 10^9$ PFU/kg/day) was intravenously injected and tipifarnib (500 μg/kg/day) was intraperitoneally injected four times, and the drugs were given to the mice without blinding. The tumor length and width were measured, and the volume was calculated according to the formula (length × width²) ÷2.

**Ex vivo model**. This study was approved by Ethics Committee of Sun Yat-sen University Cancer Center. Informed consent was obtained from all human participants. Tumor histoculture end-point staining computer image analysis (TECIA) was used to detect the ex vivo anticancer activity of M1. As reported, TECIA was an improved histoculture drug response assay[38]. Primary cancer tissue specimens from patients were dissected into small volume, and cultured. Twenty-four hours later, vehicle, M1 virus, tipifarnib, or M1/Tipi were added into the culture system; MTT assay was used to detect cell viability by the TECIA system 96 h later.

**Statistical analysis**. SigmaPlot software was used to perform the statistical analyses. Comparison between different groups was analyzed by Student's test or a one-way analysis of variance (one-way analysis of variance (ANOVA)) followed by least significant difference tests. When data could not satisfy the conditions of ANOVA, Kruskal–Wallis test was used. All error bars indicated SE unless otherwise indicated. Values of tumor volume were analyzed by repeated-measures ANOVA. Significant differences were accepted if the $P$-value was < 0.05.

**Data availability**. Gene expression data have been deposited in the GEO profiles database under the accession codes GSE99213. The authors declare that all the other data supporting the findings of this study are available within the article and its Supplementary Information files, and from the corresponding author upon reasonable request.

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

## Acknowledgements

We thank Professor Liwu Fu and Professor Xiangming Lao for the assistance in ex vivo experiments. This work was supported by National Natural Science Foundation of China (No. 81703537, No. 81573447, and No. 81773751), Research and Development Project of Applied Science and Technology of Guangdong Province, China (No. 2016B020237004), the Natural Science Foundation of Guangdong Province, China (No. 2014A030313156), the specific project of science and technology of Guangzhou (No. 201607010396), and China Postdoctoral Science Foundation (No. 2015M580761).

## Author contributions

Y.G.M., L.Y., and L.J.K. conceived and directed the project. L.J.K., G.L., L.K., and X.X. wrote the manuscript. Z.W.B. edited the manuscript. L.J.K., G.L., L.K., X.X., Z.X.K., H.J., Z.H.P., C.J., T.Y.Q., L.C.T., Y.Y.Y., H.C., L.Y., and L.X.C. performed and analysed experiments. Q.P.X., S.X.W., and H.S.M. provided key services, materials and guidance, and commented on the manuscript.

## Additional information

**Competing interests:** The authors declare no competing interests.

