## [Peer Review File · Nature Communications]

Reviewers' comments:

Reviewer #1 (Remarks to the Author):

In the manuscript entitled, "Inhibition of the Mevalonate/Farnesylation/1 RHOQ Axis Selectively Enhances M1 Virus-based Oncolysis" the authors provide convincing evidence of the antiviral role of a branch of the cholesterol biosynthesis pathway in M1 viral infection and provide some insight into the mechanism through which this occurs. Overall the manuscript is well written and has appropriately designed experiments with clear supporting data. The authors begin with showing that M1 refractory cell lines downregulate genes associated with cholesterol biosynthesis. The figure legend here does not correlate with the panels shown (A-D instead of A-G). Through a combination of siRNA and small molecule inhibitors the authors provide evidence of HMGCR and downstream FNTB as key enzymes mediating antiviral immunity in two cell lines. Farnesyl transferase has a multitude of targets for farnesylation and the authors have identified 5 candidate proteins downstream of FT that impact cell viability after siRNA knockdown. A stronger case should be made for not choosing GNG11, which shows stronger upregulation of viral proteins after siRNA knockdown than RHOQ, panel 3C. The figure caption for Figure 3 refers to TC10, for consistency in the manuscript this should be changed to RHOQ. [Page7 P2] – the authors make the claim that FT inhibitor alters the cellular location of RHOQ. This is not supported by the Supplementary Figure 4. RHOQ is found in the cytoplasm in both the vehicle and treated groups, although the expression does appear to be decreased. Next the authors show that inhibition of the FT pathway blocks interferon-related expression of the antiviral factors IRF7 and IRF3, which is arguably the most important data set for a mechanism by which this pathway contributes to antiviral immunity. The data here seem to come from only one experiment with 3 technical replicates. Biological replicates should be performed for this set of experiments to ensure reproducibility and confidence in the data. Additionally, the authors should discuss the relationship between RHOQ and its interaction with IFN- β and IRF3/7 expression in the manuscript. There is no clear connection or reasoning as to why they chose to look at this pathway or relationship other than type I interferons are expressed after PRR recognition of PAMPs. Next the authors identify numerous cell lines that show sensitivity to M1 only when combined with the FT inhibitor tipifarnib. A major limitation of this paper is the lack of investigation into in vivo efficacy of the M1/tipi-sensitive cell lines. Including animal data with tumor growth suppressed in the liver, breast, brain and bladder cancers in normal and even immunocompromised mice (which have some innate immunity or can be adoptively transferred such innate immune) would improve the scope of the study and show that the tumor growth is suppressed solely by innate immunity.

Reviewer #2 (Remarks to the Author):

This manuscript investigates the anti-viral effect of the mevalonate pathway on oncolytic virus M1 in refractory cancer cells and provides evidence for a combination strategy of targeting the mevalonate pathway for potentiating oncolytic virus therapy. The authors show that a potent inhibitor, tipifarnib, of the prenylation branch point (specifically farnesylation) of the mevalonate pathway can enhance oncolytic virotherapy both in in vitro and in vivo models.

This is an interesting manuscript that is consistent with many previous studies of the mevalonate-cholesterol pathway showing anti-viral and anti-inflammatory actions. New findings are presented for the involvement of the farnesylated GTPase RHOQ in the antiviral response. However there are a number of major weaknesses and concerns with the data presented and which is in numerous

cases is insufficient to justify the conclusions drawn. Critically, the majority of experiments shown have no more than two replicates and in some instances conclusions are drawn based on a single experiment. This combined with concerns regarding the lack of critical control experiments severely dampens enthusiasm for the publication of this work in its present form.

Major concerns:

1. In the first paragraph of the results section and figure 1 the authors infer that the refractory nature of cancer cell lines HCT-116 and SW1990 in comparison to M1 virus sensitive cell line, Hep3B is associated with the suppression of the cholesterol pathway. However, data comparing gene expression changes in Hep3B cells is not shown. It is important to show data determining whether there is a qualitative or quantitative difference in the cholesterol pathway between the sensitive and refractory cells.
2. A single contrast analysis of microarray data from infected and uninfected cells shown in Figure 1 is based on 2 biological samples and is NOT sufficient to draw any statistically significant results.
3. In the second paragraph the authors conclude: "These results confirmed that the M1 virus significantly inhibits cholesterol lipid metabolism..." Since it is known that virus infection suppresses the cholesterol pathway through the TLR- IFN response and that reduced flux in the cholesterol pathway further activates via STING-IRF3 interferon antiviral immunity –it is important for these experiments to account for or address the role of the host-interferon response over the virus in governing the MI virus response in the cell systems studied.
4. In Figure 2 data showing the specificity of the metabolic rescue experiments in refractory and a sensitive cell line (LO2) is almost completely based on n=2. Not only should more repeated independent experiments be performed but the conclusions drawn would be significantly enhanced if similar outcomes are also observed for other resistant and sensitive cell lines.
5. RNAi experiments need further clarification in the methods and results section to understand better the data presented. In the text describing Fig 2H-I it mentions no.3 siRNAi targeting FNTB. This suggests that the authors are using potentially deconvoluted siRNAs. If this is the case then all siRNAs should be shown and not cheery picking an individual. Further the pooled activity should also be shown. This needs clarification.
6. In Figure 3 conclusions are drawn on n=2 data and the main new finding in this study regarding RHOQ are based on n=1.
7. Further in Fig3 H (n=1) caution should be noted in using siRNAs above 25nM that are well known to non-specifically induce type I interferon antiviral response. Therefore while data is shown for 12.5nM all other higher concentrations are suspect for non-specific effects and requires control experiments in interferon signaling deficient cells.
8. Figure 4 G-I draws the conclusion that RHOQ exerts antiviral functions through IRF3 and IRF7 – Unfortunately the data shown for IRF3 shows a poor non-significant response. Moreover for this conclusion to be drawn further experiments are required – in particular either genetic resection of IRF3 or 7 using CRISPR technology or siRNA knock-down should be performed as critical control experiments to assess the cooperative action of tipifarnab.

Overall there is significantly more work required to support the conclusions drawn.

Response to reviewers' comments

Manuscript: NCOMMS-17-12904A

Reviewer #1:

1. The reviewer carefully found that our legend in figure 1 is incomplete. It's our mistake to sign a wrong letter to each figure legend in figure 1, and we have changed it right in the revised manuscript on page 23 and 24.

2. The reviewer suggested that we should explain why not choosing GNG11 for further verification, for siRNA knockdown of GNG11 showed stronger upregulation of viral proteins than RHOQ in figure 3C. To answer this question, we first rearranged the gene order according to the cell viability loss after gene knock-down and M1 treatment. We can see that RHOQ was the most potent gene to promote M1 virus-induced cell death, and had little impact on the cell viability when knock-down alone, while GNG11 was the fourth. NRAS knock-down alone decreased cell viability (Figure 3B).

Secondly, we further validate the 5 candidate proteins downstream of FT in figure 2C by using M1-GFP virus, which can express GFP during replication in infected cells, to quantitatively monitor the virus infection rate via flow cytometry. The result showed that both siRNA of RHOQ, GNG11, PTP4A3 and NRAS can enhance M1 infection, and the enhancement of siRHOQ was larger than other genes (Figure 3C).

Above data supported that the antiviral function of FT was mediated by 4 genes and RHOQ may play a more critical role. Therefore, we chose RHOQ for further investigation, and the antiviral mechanism of other genes would be explored in the future. These results were presented in the modified figure 3 B-C and supplementary figure 6. We have modified the text to reference these data on Page 6 line 13-17.

3. The reviewer carefully found that RHOQ is expressed as TC10 in the figure caption for Figure 3. Indeed, RHOQ is the official symbol, and TC10 is one of its aliases. For consistency, all TC10 have been changed to RHOQ in the revised manuscript.

4. We do agree with the reviewer that in both the vehicle and treated groups, RHOQ were located in the cytoplasm. Farnesyl transferase inhibitor (FTI) did not change the intracellular location of RHOQ, but induce RHOQ to be a dispersion state rather than an aggregating state. By measuring the fluorescence of RHOQ in HCT-116, we found that FTI can either decrease its intensity or disperse it, for the frequency of signal after tipifarnib treatment increased (Supplementary Figure 8B). So the expression in page7 line17 should be changed from “FT inhibitor tipifarnib apparently alter its intracellular location in HCT-116 cells” to “FT inhibitor tipifarnib apparently induced RHOQ dispersion in HCT-116 cells”. This analysis data was presented in supplementary figure 8B.

5. The reviewer pointed out that the conclusion was mainly based on three technical replicates in Figure 4. To further validate our results, we replicated the data in the whole figure for at least 3 biological replicates. In accordance, pretreatment of tipifarnib can significantly block IFN- β -induced IRF7, IFN- β and MDA5 mRNA expression (Supplementary Figure 9). Tipifarnib obviously enhanced M1 virus replication, and blocked M1 virus-induced IRF3 and IRF7 expression (Supplementary Figure 10) as expected. Further silencing IRF3 and IRF7 can increase M1 virus infection (Supplementary Figure 11). The above data prove that IRF3 and IRF7 mediated the antiviral function of protein farnesylation. So, we proposed that RHOQ regulated by protein farnesylation may exert its antiviral role through IRF3 and IRF7. However, RHOQ did not suppress IRF3 and IRF7 expressions (Supplementary Figure 12). This result indicated that RHOQ did not mediate its antiviral effect through IRF3 and IRF7.

Therefore, to further find out the antiviral mechanism of RHOQ, we use microarray to analyze the changes of expression profiles after RHOQ knock-down. GO analysis showed that RHOQ depletion-induced differences were mainly focused on four aspects (Figure 4 A-B) comparing to negative control. Molecular function analysis indicated that genes related to Rab GTPase and ATP-mediated membrane transporter system (vacuolar ATPase and ATP-binding cassette transporters) were down-regulated after RHOQ knock-down (Figure 4 C-D). The above data implicated that RHOQ involved in the process of intracellular membrane and transportation, which may mediate the antiviral effect of RHOQ.

Above data were allocated in supplementary 9-12, and Figure 4. We have also revised the text to reference these data in page 8 and page 9 lines 1-6, and the discussion focusing about the relationship between viruses and intracellular membrane trafficking in page 12 lines 1-23.

6. The reviewer questioned that “no clear connection or reasoning as to why they chose to look at this pathway or relationship other than type I interferons are expressed after PRR recognition of PAMP”. The reason why we focused on IRF3/7 rather than IFN- β was that we have proved that resistant cells did not exploit type I IFNs to establish the antiviral state against M1 virus. While we used the inhibitor of JAK-1 to block the pathway activated by IFN- α and IFN- β , the cytopathic effect caused by M1 virus was not obviously promoted in resistant cells (Additional Figure 1- **[Redacted]**). Moreover, IRF3 and IRF7 as transcription factors regulate a series of antiviral factors directly, which can circumvent the induction of IFN β (Honda, K, et al. *Nat Rev Immunol*, 6(9): 644-658, 2006). For these reasons, we mainly focused on the antiviral effect of IRF3 and IRF7 rather than IFN- β .

7. The reviewer suggested that the tipifarnib plus M1 virus-sensitive cell lines in figure 5 A were needed to confirm *in vivo*. As it is showed in Figure 5 A, besides pancreatic and colorectal cancer cell lines, liver cancer cell lines HepG2 and Huh-7 were sensitive to the combination therapy than other cancer cell lines. However, we failed to establish xenograft models using these two cell lines in nude mice. Nevertheless, we utilized a more clinically relevant model to test the combination effect of M1 virus and tipifarnib. Tested cancer samples were from 10 liver cancer patients and 9 colon cancer patients. *Ex vivo* results proved that M1 virus plus tipifarnib significantly enhanced both liver and colorectal cancer cell death (Figure 6 G-H).

We have enclosed this data in Figure 6 G-H. And we have also revised the text to reference these new figures in page 10 lines 27-28, page 11lines 1-2.

Reviewer #2:

1. The reviewer suggested that it is important to show data determining whether there was a qualitative or quantitative difference in the cholesterol pathway between the sensitive and refractory cells. In terms of this critical concern, we performed expression profiling to detect lipid-related genes in M1-sensitive cell line Hep3B. In sensitive cancer cells line Hep3B, the metabolism of four lipids were dramatically changed after M1 virus infection, as the unchanged gene group was obviously decreased comparing to the refractory cell lines. This may be due to exponential virus replication consumption in sensitive cancer cells. However, the genes in cholesterol metabolism pathway did not show a similar pattern as HCT-116 and SW1990 (Supplementary Figure 1).

We have enclosed these new data in Supplementary Figure 1. We have also modified the text to reference these new data on page 3 lines 20-23.

2. The reviewer raised the question that present data of microarrays were not sufficient to draw statistically significant results. We do agree that our microarray data is not sufficient to draw any statistically significant results. However, we analyze the microarray data, including HCT-116, SW1990 and Hep3B, with the same standards. Moreover, the hints from data analysis were further confirmed by qRT-PCR. The qRT-PCR data was from three biological replicates.

3. We do agree with the point that M1 virus infection stimulates TLR-IFN response to suppress cholesterol pathway, which in turn further activates interferon antiviral immunity. It's reported that human cytomegalovirus infection stimulated IFN β /IFN α secretion, which suppressed sterol biosynthesis via activating the IFNAR1 pathway (Blanc, M, et al. *PLoS Biol*, 9(3): p. e1000598, 2011). Moreover, we have proved that M1 virus infection induced the IRF3 and IRF7 mRNA expressions (Supplementary Figure 10). So down-regulation of cholesterol genes were not directly induced by M1 virus, but governed by the host-interferon response. The expression in page 4 lines 1-4 have changed from "These results confirmed that the M1 virus significantly inhibits cholesterol lipid metabolism in refractory cancer cells" to "These results confirmed that Cholesterol lipid metabolism pathway is significantly suppressed after M1 virus infection in refractory cancer cells."

4. The reviewer suggested that the metabolic rescue experiments in figure 2 F should be repeated more than twice and in more cell lines including refractory and sensitive to confirm the inhibition of mevalonate pathway during M1 virus replication. To address this concern, we used fluvastatin, an inhibitor of HMG-CoA reductase, to suppress mevalonate pathway, and infected HCT-116 (refractory), SW1990 (refractory) and Hep3B (sensitive) cells with M1-GFP virus, which can express GFP during virus replication in infected cells. Simultaneously downstream metabolite FPP was added back to the culture system to rescue the pathway. By detecting infection rate of M1 virus with flow cytometry, we found that fluvastatin significantly enhanced infection rate of M1 virus, and replenishment of FPP can reverse M1 virus infection in refractory cancer cells HCT-116 and SW1990 (Figure 2 F). In sensitive Hep3B cells, fluvastatin slightly inhibited M1 virus infection, and FPP did not affect M1 virus infection rate (Supplementary Figure 3). All the experiments were repeated more than three times. These results are consistent with our hypothesis that mevalonate pathway inhibits M1 virus.

We have enclosed these new data in Figure 2 F and Supplementary Figure 3.

5. To make the RNAi experiments method more clear, we have detailed the RNA interference assay in method section “RNA interference. siRNAs were synthesized by Ribobio (Guangzhou, China). Cells are seeded in proper density and incubated for 24 hours, then siRNAs were transfected with Lipofectamine RNAiMAX (13778-150, Thermo Fisher, Rockford, IL, USA) and OPTI-MEM (31985070, Thermo Fisher) in concentration recommended by instruction. 24 hours later, supernatants were removed and changed to new medium. Then the cells were treated according to different experiments. The concentration of siRNA for different genes are as follows: HMGCR (25 nM), FNTB (3 nM for HCT-116, 10 nM for SW1990 and L02), SQLE (40 nM), PGGT1B (40 nM), RABGGTB (40 nM), RHOQ (25 nM for HCT-116, 40 nM for SW1990 and L02), GNG11 (25 nM), PTP4A3 (25 nM), PTGIR (25 nM), NRAS (25 nM), IRF3 (40 nM), IRF7 (40 nM).”

Furthermore, for FNTB knocking-down experiment in figure 2 I-J, we only used No.3 siRNA. To be more objective, we used three siRNA and the mix of three siRNA targeting FNTB to test whether FNTB knock-down promote M1virus-induced cell apoptosis as the reviewer recommended. The results showed that silencing of

FNTB elevated M1 virus-induced cell apoptosis both in HCT-116 and SW1990, in which NO.2 siRNA alone obviously induced cell death and promoted M1 virus-induced cell apoptosis slightly, NO.3 and Mix siRNA showed the most potent capacity to elevate M1 virus-induced cell apoptosis (Supplementary Figure 4)..

We have enclosed these new data in Supplementary Figure 4. We have also modified the text to reference these new data on page 5 lines 17-19.

6. We agree with that the conclusions for figure 3 were based on limited replicates. So we replicated enough times to ensure the credibility of our results.

For figure 3C, we further validate the 5 candidate proteins downstream of FT by M1-GFP virus to quantitatively monitor the virus infection rate by flow cytometry for three biological replicates. The result showed that knocking-down of GNG11, RHOQ, PTP4A3 and NRAS can specifically enhance M1 infection, and the average enhancement of siRHOQ was larger than other genes (Figure 3C).

For figure D-F, we used one more cancer cell line SW1990 to confirm the antiviral function of RHOQ. By knocking-down RHOQ in SW1990 cell, M1 virus-induced cytopathic effect, M1 virus infection rate and virus titer were significantly enhanced (Supplementary Figure 6, Figure 3 F). Individual experiments here replicated for at least three times.

For figure 3 G-I, three biological replicates have finished, and the results were consistent with previous data. RHOQ was highly expressed in M1 virus-refractory cancer cell lines HCT-116 and Capan-1, but was deficient in the M1 virus-sensitive Hep3B and T24 cell lines (Figure 3 G). Moreover, the FT inhibitor tipifarnib notably increased the M1 virus-induced loss of viability in RHOQ normal cells in a dose-dependent manner but not in RHOQ-deleted cells (Figure 3 H). These results further verified the antiviral effect of RHOQ. Furthermore, we confirmed that RHOQ expression was decreased after FT inhibition or combining with the M1 virus via western blotting for three times (Figure 3 I).

7. We agree that siRNA can induce nonspecific interferon antiviral response, however, it is reported that this antiviral response is activated when the siRNA is

more than 30 bp, and with some specific base sequence that more easily to induce interferon response. SiRNA with 19-21 bp is proper for gene knock-down experiment to effectively function but circumvent non-specific effect (Hornung, V., et al. *Nat Med*, 11(3): p. 263-70, 2005; Persengiev, S.P., et al. *RNA*, 10(1): p. 12-8, 2004). For our experiments, the siRNAs were 19 bp long, and circumvented all the sequences that may induce interferon response. What's more, a negative control which employed scramble siRNA at the same concentration was set for all the siRNA experiments. So, we believed that nonspecific interferon antiviral response was well controlled in our experiments.

8. Here, we summarized up the evidence about the antiviral mechanism of RHOQ: As mentioned in the text, to explore the antiviral mechanism of RHOQ, we firstly focused on antiviral innate immunity because it was reported that protein farnesylation inhibited inflammatory factors. In accordance, pretreatment of tipifarnib can significantly block IFN- β -induced IRF7, IFN- β and MDA5 mRNA expression (Supplementary Figure 9). Moreover, tipifarnib obviously enhanced M1 virus replication, and blocked M1 virus-induced IRF3 and IRF7 expression (Supplementary Figure 10) as expected. Further silencing IRF3 and IRF7 can increase M1 virus infection (Supplementary Figure 10). The above data prove that IRF3 and IRF7 mediated the antiviral function of protein farnesylation. So, we propose that RHOQ regulated by protein farnesylation may exert its antiviral role through IRF3 and IRF7. However, RHOQ did not suppress IRF3 and IRF7 expressions (Supplementary Figure 12). This result indicated that RHOQ did not mediate its antiviral effect through IRF3 and IRF7.

Therefore, to further find out the antiviral mechanism of RHOQ, we used microarray to analyze the changes of expression profiles after RHOQ knock-down. GO analysis showed that RHOQ depletion-induced differences were mainly focused on four aspects (Figure 4 A-B) comparing to negative control. Molecular function analysis indicated that genes related to Rab GTPase and ATP-mediated membrane transporter system (vacuolar ATPase and ATP-binding cassette transporters) was down-regulated after RHOQ knocking-down (Figure 4 C-D). The above data implicated that RHOQ involved in the process of intracellular membrane and transportation, which may mediate the antiviral effect of RHOQ.

Above data were allocated in supplementary 9-12, and Figure 4. We have also revised the text to reference these data in page 8 and page 9 lines 1-6, and the discussion focusing about the relationship between viruses and intracellular membrane trafficking in page 12 lines 1-23.

[Redacted]

Reviewers' comments:

Reviewer #1 (Remarks to the Author):

In general I am satisfied with the authors' responses to my comments and concerns. I believe they have made substantial attempts to address all issues raised.

Reviewer #2 (Remarks to the Author):

The revised manuscript has addressed a number of concerns. However there remain issues around how the data is analysed and discussed.

Firstly the microarray data is insufficient to draw any statistically meaningful conclusion -- however there are many studies to suggest that the reduction of the cholesterol pathway upon infection is as anticipated and perhaps this should be corroborated by the RT-PCR experiments - to be shown in Figure 1.

Secondly results should be shown as SE (standard error) of the mean and NOT SD (standard deviation).

Thirdly different statistical tests are performed -- this needs to be consistent using the same test throughout. The concern here is the cherry picking of a statistical tests that show significance.

Response to reviewers' comments

Reviewer #2

Concern 1 :

We appreciate very much for the reviewer's comment. In our research, the microarray data indicated that mevalonate-cholesterol biosynthesis pathway was down-regulated after M1 virus infection. The phenomenon was corroborated in two cell lines (HCT-116 and SW1990) by detecting five genes in this pathway with qRT-PCR method, just as the reviewer suggested. We did not further explore the mechanism why this pathway was decreased, but as the reviewer mentioned, many viruses infection suppressed the cholesterol biosynthesis pathway by inducing IFN- β excretion, such as HCMV (Mathieu Blanc, *Plos Biology*, 2011) and MHV-68 (Autumn G. York, *Cell*, 2015). It is probably that M1 virus infection suppressed this pathway by inducing antiviral transcription factor or IFN- β . And in our research, we found that suppressing mevalonate/farnesylation pathway can enhance the replication and oncolytic capacity of M1 virus.

Concern 2 :

The reviewer suggested that results should be shown as standard error (SE) of the mean rather than standard deviation (SD). We do agree that SE should be used, for SD is used to describe the variation of single value, and SE represents the accuracy of the existing data to the true value. All the quantitative data have been revised to represented as mean \pm SE.

Concern 3 :

We agree that the statistical test methods were not consistent. Therefore, we reanalysed all the data using proper statistical tests. Comparisons between two groups were analysed by Student's-t-test, and for more than two groups, one-way ANOVA followed by least significant difference tests (*LSD-t*) were performed. When data could not satisfy the conditions of one-way ANOVA, Kruskal-Wallis test was used. Values of tumor volume were analysed by repeated-measures one-way ANOVA. With the above analysis, most of the results were consistent with previous statistical results, except for figure 2L and supplementary figure 11.

REVIEWERS' COMMENTS:

Reviewer #2 (Remarks to the Author):

The response of the authors is satisfactory.